# Non-Genetically Encoded Epitopes Are Relevant Targets in Autoimmune Diabetes

**DOI:** 10.3390/biomedicines9020202

**Published:** 2021-02-17

**Authors:** Hai Nguyen, Perrin Guyer, Ruth A. Ettinger, Eddie A. James

**Affiliations:** Translational Research Program, Benaroya Research Institute at Virginia Mason, Seattle, WA 98101, USA; hnguyen@benaroyaresearch.org (H.N.); pguyer@benaroyaresearch.org (P.G.); rettinger@benaroyaresearch.org (R.A.E.)

**Keywords:** type 1 diabetes, neo-antigen, post-translational modification, T cell, selection

## Abstract

Islet antigen reactive T cells play a key role in promoting beta cell destruction in type 1 diabetes (T1D). Self-reactive T cells are typically deleted through negative selection in the thymus or deviated to a regulatory phenotype. Nevertheless, those processes are imperfect such that even healthy individuals have a reservoir of potentially autoreactive T cells. What remains less clear is how tolerance is lost to insulin and other beta cell specific antigens. Islet autoantibodies, the best predictor of disease risk, are known to recognize classical antigens such as proinsulin, GAD65, IA-2, and ZnT8. These antibodies are thought to be supported by the expansion of autoreactive CD4^+^ T cells that recognize these same antigenic targets. However, recent studies have identified new classes of non-genetically encoded epitopes that may reflect crucial gaps in central and peripheral tolerance. Notably, some of these specificities, including epitopes from enzymatically post-translationally modified antigens and hybrid insulin peptides, are present at relatively high frequencies in the peripheral blood of patients with T1D. We conclude that CD4^+^ T cells that recognize non-genetically encoded epitopes are likely to make an important contribution to the progression of islet autoimmunity in T1D. We further propose that these classes of neo-epitopes should be considered as possible targets for strategies to induce antigen specific tolerance.

## 1. Introduction

Type 1 diabetes (T1D) is an autoimmune disease in which insulin-producing β cells are destroyed, leading to lifelong insulin deficiency [1,2]. Autoantibodies against specific, biochemically defined antigens—including insulin, glutamic acid decarboxylase (GAD65), protein tyrosine phosphatase (IA-2), and zinc transporter 8 (ZnT8)—accumulate prior to disease onset [3]. Although not directly implicated in disease etiology, these autoantibodies represent the best indicators of disease risk and their formation reflects loss of tolerance toward beta cell specific antigens [4]. The appearance of high-affinity antibodies implies that self-reactive CD4^+^ T cell responses are present to provide help to autoreactive B cells. Indeed, both CD4^+^ and CD8^+^ T cells from peripheral blood have been shown to recognize a diverse array of epitopes derived from insulin and other beta cell antigens such as GAD65, IA-2, and ZnT8 [5,6,7,8,9,10]. Furthermore, auto-reactive T cells with some of these antigen specificities have been shown to infiltrate pancreatic islets [11,12]. In particular, the genetics of T1D implicate CD4^+^ T cell responses, as a crucial factor in disease development, in that disease risk is associated with susceptible human leukocyte antigen (HLA) class II alleles. Specifically, the *DRB1*03:01-DQA1*05:01-DQB1*02:01* and *DRB1*04:01-DQA1*03:01-DQB1*03:02* haplotypes (alone or in combination) exhibit a major proportion of the overall genetic risk [13]. It is notable that HLA proteins are the most polymorphic genes in the human genome and have been associated with more diseases than any other region [14,15]. These polymorphisms result in structural differences in the size and shape of binding pockets along the peptide-binding groove, conferring allele specific motifs, which dictate the antigenic peptides that can be presented to T cells by antigen presenting cells [16]. Experimental evidence reveals HLA specific differences in T cell receptor (TCR) distribution (showing that HLA genotype shapes the T cell repertoire) and, notably, several of the polymorphisms that exhibit the strongest impact on TCR gene segment usage align with key structural features of DQB1*02:01 (HLA-DQB1 p57), DQB1*03:02 (HLA-DQB1 p57) and DRB1*04:01 (HLA-DRB1 p70, p71, and p73) [17]. Therefore, it can be surmised that T1D susceptible HLA class II proteins contribute to disease by facilitating the selection of a potentially autoreactive T cell repertoire. Although some self-reactive T cells are specific for conventional self-epitopes, it is increasingly appreciated that T cells respond to non-genetically encoded neo-epitopes, including enzymatically modified antigens, hybrid peptides, splice variant peptides, and defective ribosomal products (DRiP) [18,19,20]. As we delineate further in the sections that follow, the formation of non-genetically encoded antigens and epitopes represents a significant challenge to self-tolerance. Their formation represents a nexus between the natural secretory function of pancreatic β cells and inflammatory and environmental stresses. The result is the generation of self-proteins with non-templated sequence changes that alter their immunogenicity. To support the relevance of such non-genetically encoded epitopes, this review will provide an overview of central and peripheral tolerance, discuss different classes of neo-epitopes that have been implicated in T1D, and highlight their potential as targets of antigen specific tolerance induction.

## 2. Central and Peripheral Tolerance

An almost innumerable variety of unique TCRs can be formed through V(D)J recombination, but only a subset of these acquire a functional T cell receptor that will be adequate for positive selection in the cortex [21]. It is estimated that a significant proportion of positively selected receptors exhibit potentially dangerous self-antigen recognition [22]. However, tolerance mechanisms serve to eliminate and mitigate T cell self-reactivity (Figure 1). Central T cell self-tolerance is enforced in the thymic medulla, where the autoimmune regulator protein (AIRE) allows medullary thymic epithelial cells to express tissue specific antigens [23,24]. Of specific relevance to T1D, there is detectable expression of insulin in medullary thymic cells [25]. T cells with excessive self-recognition (e.g., those that recognize AIRE associated self-proteins), are largely eliminated through the process known as negative selection or clonal deletion [22]. There is some evidence to suggest that immature T cells can also undergo receptor editing through secondary rearrangement of the TCR α-chain [26]. In this scenario, T cells with excessive self-recognition may escape negative selection by replacing or diluting their TCR α-chain with a second α-chain that is less self-reactive [22]. In spite of these mechanisms, negative selection is imperfect such that the repertoire of healthy subjects with susceptible HLA haplotypes includes autoreactive T cells [23,27]. Indeed, it has been hypothesized that autoimmune susceptible HLA proteins may exhibit suboptimal or unstable presentation of particular self-peptides, leading to inefficient clonal deletion during negative selection [28,29]. Evidence from multiple studies suggests that thymic expression of a particular self-antigen promotes tolerance and lack of expression impairs tolerance to that antigen. For example, enforced expression of hen egg lysozyme (HEL) led to a reduction in the number of HEL specific T cells in the thymus and periphery and a reduced proliferative response to HEL [30]. In human T1D, the IDDM2 locus maps to a variable number of tandem repeats (VNTR) minisatellite upstream of the insulin gene, for which the protective VNTR genotype is associated with 2- to 3-fold higher levels of thymic insulin expression than the susceptible genotype [31]. Furthermore, the susceptible VNTR genotype was correlated with higher peripheral frequencies of insulin specific T cells [32].

As a complement to central tolerance, T cell self-reactivity is further tuned in the periphery through multiple layers of intrinsic and extrinsic regulation [22,33]. In some cases, a potentially self-reactive T cell may never encounter adequate levels of its cognate antigen, maintaining a state of immunologic ignorance. However, given the necessary cues, epitope specific T cells will be drawn into secondary lymphoid organs or tissue and activated. Upon engagement of a TCR by its cognate epitope, there is evidence for intrinsic regulation through modulated expression of CD5, which is induced in proportion to the strength of TCR self-reactivity [34,35]. Adequate unresponsiveness to self is further reinforced by the activity of inhibitory receptors such as cytotoxic T-lymphocyte-associated antigen 4 (CTLA-4), which disrupts co-stimulatory ligands B7-1 and B7-2 to induce anergy [33]. In a complementary fashion, PD-1 transduces an inhibitory signal when engaged simultaneously with the TCR which limits self-reactive T cell proliferation and cytokine production [36]. Notably, the PD-1–PD-L1 pathway regulates both the initiation and progression of autoimmune diabetes in nonobese diabetic (NOD) mice, in which the presence of PD-L1 on inflamed islets implies a direct protective effect within disease proximal tissue [37]. Peripheral tolerance is also maintained by extrinsic regulation through the activity of populations of regulatory T cells and tolerogenic dendritic cells (DCs). Tolerogenic DCs present peptides in the context of low levels of co-stimulation leading to anergy or deletion rather than activation [33]. Tregs suppress effector T cell responses through a variety of mechanisms, including secretion of IL-10 and TGF-β, adenosine signaling, and CTLA-4 surface expression [38]. These mechanisms are interrelated in that Tregs have suppressive interactions with antigen presenting cells and tolerogenic DCs may promote the induction of peripherally derived Tregs [33]. Nevertheless, peripheral tolerance mechanisms are diminished and can be overcome in scenarios where infection or local inflammation creates a milieu in which danger signals direct DCs toward an immune priming phenotype and inflammatory cytokines oppose regulation and support effector function [39].

## 3. Non-Genetically Encoded Neo-Antigens and Epitopes

The inflammatory environment that diminishes peripheral regulation also promotes the formation of neo-antigens (Figure 2). Inflammatory cytokines, reactive oxygen species, and other stresses (Figure 2a) promote post-translational modifications and alternative mRNA splicing of various β cell proteins (Figure 2b) through several parallel stress related processes that impact the amino acid composition (Figure 2c) or alter mRNA (Figure 2d) available for protein translation [19,40]. These processes generate neo-epitope sequences that are not genetically encoded, posing a formidable challenge to tolerance. As such, neo-epitopes are likely to be underrepresented in the thymus (undermining central tolerance) and essentially absent in healthy tissue (undermining peripheral tolerance). This theme has been previously described as ‘autoantigenesis’, a process to indicate how proteins acquire enzymatic and biochemical modifications over the progression of disease and stimulate waves of B cell and T cell autoimmunity [21]. This phenomenon is recognized as being relevant for a number of autoimmune diseases, including multiple sclerosis, rheumatoid arthritis, systemic lupus erythematosus (SLE) [20,41], but our focus will be on T1D. For T1D, pancreatic β cells are the focal point of disease and their unique biology can drive the emergence of tissue specific antigens and epitopes. Because of their role as professional secretory cells, β cells carry out extremely high levels of protein translation to synthesize and process adequate stores of mature insulin that can be released as needed to maintain normal glucose levels [42,43]. The environmental trigger that precipitates the loss of tolerance in T1D is unknown, but it is notable that processes shown to generate neo-sequences can be easily tied to suspected triggers of T1D. For example, enteroviral infection (e.g., with Coxsackie virus, which accelerates NOD diabetes) is a known trigger for inflammatory cytokines, such as interferon-γ [44]. The anti-viral immune response is thought to synergize with pre-existing autoreactive T cells to precipitate destruction of β cells [45]. The secretion of pro-inflammatory cytokines by infiltrating immune cells [46,47] sets off a cascade of events that includes HLA class I upregulation, expression of pro-apoptotic proteins, mitochondrial dysfunction and endoplasmic reticulum (ER) stress [48,49]. These events converge to elicit important changes to both the transcriptional profiles and the proteome of β cells [50,51,52,53]. Beyond viral infection, there are other plausible triggers, including metabolic stress [54] and exposure to chemicals [55,56]. As will be discussed below, one common downstream component of these putative environmental triggers is increased cytosolic calcium, leading to increased activity of the peptidyl arginine deiminase (PAD) enzymes responsible for protein citrullination and tissue transglutaminase (tTG) enzyme responsible for protein deamidation [40]. Complementary mechanisms lead to the increased formation and presentation of splice variant peptides and DRiP derived peptides.

### 3.1. Insulin Derived Neo-Antigens

Among the antigens known to be targeted by antibodies and T cells in T1D, a strong case can be made for the importance of insulin. Notably, insulin is uniquely produced by pancreatic β cells. Natural history studies show that insulin specific antibodies often appear first in at risk subjects, followed by glutamic acid decarboxylase specific antibodies and other specificities [57]. Insulin specific responses have been shown to play a central role in the NOD mouse model, in that responses to insulin B 9–23 are crucial for disease development [58]. Studies of human T1D affirm the relevance of the insulin B 9-23 epitope and additional insulin epitopes, such as proinsulin 76–90, for which there are comparatively higher T cell frequencies in subjects with DRB1*04:01 haplotypes [59]. As elaborated above, compromised tolerance to insulin can be partially explained by the susceptible VNTR genotype, which results in lower thymic insulin expression and (presumably) impaired clonal deletion [31]. However, decades of study make it clear that biological processes can alter the proinsulin transcript or modify proinsulin and insulin proteins at key amino acids, thereby generating non-genetically encoded insulin epitopes that pose a tolerogenic challenge [19]. We will individually discuss the most important of these processes (insulin oxidation, formation of hybrid insulin peptides, and defective ribosomal products of proinsulin) in the subsections that follow.

#### 3.1.1. Insulin Oxidation

The first report of a disease relevant non-genetically encoded neo-epitope in T1D was the observation by Mannering et al. that an insulin A-chain epitope recognized by human T cells is post-translationally modified through oxidation in a way that alters immune recognition [60]. Specifically, this work demonstrated that the immunogenicity of this epitope was dependent on the formation of a vicinal disulfide bond that altered TCR recognition but had no impact on the HLA binding of the peptide [60]. Work that is more recent demonstrated that neo-epitopes formed through the oxidation of insulin are recognized by autoantibodies in subjects with T1D and at risk subjects [61]. Oxidized insulin antibodies were more common than native insulin autoantibodies and allowed discrimination between children who progressed to develop T1D and children who did not progress [62]. These antibodies have been further refined as a biomarker, improving the prediction of diabetes risk in children [63]. Carbonylation is a distinct oxidative modification that has been reported as having potential significance in cellular stress and insulin signaling, but its immunologic significance remains to be elucidated [64]. In any case, given that oxidative stress is increased in at risk individuals and subjects with T1D [65], it is plausible that a break in tolerance towards oxidized insulin could precede the loss of tolerance to wild type insulin. However, it remains to be determined whether T cells and antibodies respond to any of the same oxidized determinants. Oxidized low density lipoproteins represent another potential source of neo-epitopes. Various oxidized and other modified LDLs are thought to be formed as a consequence of the hyperglycemia and dyslipemia that occurs as beta cell function decreases [66].

#### 3.1.2. Hybrid Insulin Peptides

The formation of hybrid insulin peptides (HIPs) is a novel form of posttranslational modification that is specific to the secretory granules of pancreatic β cells [67]. Hybrid formation involves a covalent linkage of insulin fragments with each other or with other protein fragments from separate parent molecules. The precise mechanism that promotes HIP formation has yet to be elucidated, but the result is covalent fusion of insulin fragments with other secretory granule peptides-creating neo-epitopes, which are by definition not genomically templated. HIPs were first observed in murine insulinoma cells and studied in NOD mice and were shown to be potent ligands for T cells that were isolated from islet infiltrates [68]. The HIPs with the most established relevance contain fragments of the insulin B chain, but responses to HIPs that contain other insulin fragments have also been reported [69]. Subsequent studies have confirmed the recognition of HIPs in human T1D and demonstrated that HIPs are recognized with much higher affinity than the corresponding non-hybrid sequences [70]. To some degree this increased recognition can be attributed to increased stability of binding to diabetes susceptible HLA class II proteins [69]. Importantly, HIP reactive T cells were documented among human islet infiltrating T cell lines [12]. Notably, administration of a TCR-like monoclonal antibody designed to interrupt the activation of HIP reactive T cells delayed the development of diabetes in NOD mice [71].

#### 3.1.3. DRiP

Another insulin related neo-epitope that has been reported is an insulin DRiP that is formed through translation of an alternative open reading frame within human insulin mRNA [53]. The N-terminal peptide of this nonconventional polypeptide is recognized by cytotoxic CD8^+^ T cells. T cells specific for this non-standard epitope were shown to be present at similar frequencies to those that recognize the well-established preproinsulin 15–24 epitope and exhibited a diverse range of surface phenotypes [72]. Paralleling processes that have been observed during tumor development, high demand for insulin that is accompanied by inflammatory stress may enhance the generation of aberrant insulin polypeptides formed by nonstandard translational initiation. The resulting polypeptide products could span normally untranslated regions or be translated in an alternative reading frame, which would consequently not share any sequence identity with the canonical translation product [53]. Given that similar nonstandard translation may not be expected to occur in the thymus, tolerance to such DRiP peptides could be substantially impaired.

### 3.2. Enzymatically Generated Neo-Antigens

Enzymatically modified peptides represent another important class of non-genetically encoded epitopes. As discussed above, T1D is associated with distinct HLA risk haplotypes, most notably *DRB1*03:01-DQA1*05:01-DRB1*02:01* and *DRB1*04:01-DQA1*03:01-DQB1*03:02*. These susceptible HLA class II alleles have been shown to favor the presentation of enzymatically modified neo-epitope peptides. In particular, the conversion of arginine to citrulline by PAD enzymes has been strongly linked to autoimmunity [73]. HLA-DRB1*04:01 presents citrullinated peptides as neo-epitopes because conversion of a native arginine to citrulline at specific residues dramatically improves binding to the extent that peptides that are incapable of binding in their unmodified form can exhibit moderate or even high affinity binding in their citrullinated form [74,75,76]. The recognition of citrullinated self-antigens has long been recognized as central to disease development for seropositive rheumatoid arthritis [77]. However, recent studies demonstrate that citrullinated *β* cell antigens (including GAD65, GRP78, and IAPP) are recognized by autoreactive T cells and autoantibodies [40,51,78,79]. Analogously, the HLA-DQ alleles that confer susceptibility to T1D exhibit a strong preference for presenting deamidated peptides because of a key structural feature. The T1D susceptible HLA-DQ proteins lack a canonical Asp residue at position 57 of the DQ beta chain that, when present, forms a salt bridge with an arginine at position 76 of the DQ alpha chain [80,81]. The docking of peptides with an acidic amino acid can stabilize the unpaired arginine, greatly enhancing the binding of negatively charged peptides [82,83]. This creates a binding pocket with a strong preference for negatively charged amino acids. The phenomenon of increased immune recognition of deamidated peptides formed through the activity of tTG was first demonstrated for gluten-derived peptides in the context of celiac disease [84]. However, recent studies demonstrate immune recognition of deamidated *β* cell antigens, including insulin, GAD65, and IA-2 [40,78,83]. Importantly, these enzymatically modified peptides were shown to be more immunogenic than their unmodified counterparts [40,78,79].

Another important line of evidence supports the relevance of these modifications to T1D. The activity of PAD and tTG enzymes have been shown to be increased in both human and murine beta cells through ER and inflammatory stress, leading to increased immune recognition of stressed *β* cells [40,50,51]. Although it is unclear whether the same enzymatic modifications can occur in the thymus, the relevance of citrullinated and deamidated peptides in multiple autoimmune diseases does suggest that tolerance to these non-genetically template epitopes is impaired. In light of this, there is a growing appreciation that enzymatically modified epitopes and antigens are an important facet of autoreactive responses and could reflect an important means of circumventing tolerance.

### 3.3. Neo-antigens Generated through Alternative Splicing

Another important class of epitopes and antigens are those generated through alternative splicing. Such alternatively spliced (AS) proteins are genetically encoded, but their unique splice junctions represent sequences that are confined to specific tissues, and in some cases associated with cellular stress. Ongoing work in this area has been increasingly definitive. Human islet studies show that proinflammatory cytokines can increase alternative splicing in *β* cells [85]). Specifically, interferon-α appears to play a major role in altering the splicing landscape, as demonstrated by a recent multi-omics study, which probed alternative splicing in stressed beta cells [86]. Another published study demonstrated that isoforms of IGRP are present in the pancreas but rarely detected in the thymus [87]. A deep peptidomic and transcriptomic analysis revealed that epitopes derived from splice variants of known β cell proteins and novel targets (e.g., GAD65, Secretogranin V, CCNI-008, IAPP, and Phogrin) are recognized by self-reactive T cells [52]. Although currently published work is limited to CD8^+^ T cells, ongoing work seeks to verify that CD4^+^ T cells recognize this class of antigenic targets. Notably, T cells specific for such epitopes were shown to be present within the pancreatic infiltrates of autoantibody positive donors and donors with T1D [52]. This observation affirms the relevance of these specificities as a potential therapeutic target. 

## 4. Inducing Antigen Specific Tolerance Using Peptide Neo-Epitopes

Conceptually, induction of antigen specific tolerance provides an attractive treatment option for T1D. Several strategies have been attempted and/or are currently under development to elicit tolerance to self-antigens (Table 1). Direct delivery of the PPI C19-A3 peptide elicited IL-10 secretion, presumably by antigen specific T cells [88,89]. Furthermore, the treatment showed a good safety profile and was associated with decreased insulin use and retention of residual c-peptide. This shows that with appropriate dosing and timing, free peptides can have a tolerogenic effect. HLA-peptide complexes consisting of immunodominant peptides bound to the high risk DRB1*04:01 and DRB1*03:01 proteins were shown to elicit Tr1-like cells and a broader regulatory network [90]. This suggests that engaging self-reactive TCRs in the absence of co-stimulation has multi-faceted effects that promote tolerance. Administration of either peptide or HLA-peptide coated nanoparticles resulted in reduced T cell proliferation and the induction of tolerogenic DCs [91,92]. These positive effects of peptide coated nanoparticles are attributed to their ability to mimic apoptotic bodies, whereas HLA-peptide coated nanoparticles are intended to directly engage TCR and reprogram self-reactive T cells [93]. Transfection with DNA or mRNA encoding self-peptide or a self-antigen provides a potentially powerful strategy [94,95]. Again, the goal of these approaches would be presentation of peptides to T cells in the absence of inflammatory signals. As an alternative to indirect strategies, tolerogenic DCs can be directly induced through treatment with soluble inhibitors of NF-κB and pulsed with relevant epitopes [96,97]. Murine and in vitro studies support the potential efficacy of this approach and early human trials have demonstrated its safety. Soluble antigen arrays represent a novel approach designed to deliver antigenic peptides or proteins in a multivalent form in a manner that ameliorates the risk of anaphylaxis [98]. Early studies suggest that this platform retains the positive effects of treatment with free peptides.

Successful implementation of these approaches will require adequate knowledge of disease relevant antigens and epitopes. Although key knowledge gaps remain [5], epitope knowledge in T1D has now progressed to the point that an adequate number of disease relevant antigens and epitopes are available to attempt tolerance induction through each of these emerging strategies. As summarized in the preceding paragraphs, this knowledge includes a growing array of non-genetically encoded beta cell antigens and peripherally generated epitopes. The common thread for each of these antigens and epitopes is their potential absence within the thymus and healthy tissue and links between their generation and various inflammatory processes that have demonstrated relevance in pancreatic *β* cells. Although the level of support for each individual epitope may vary [5], in total the evidence is difficult to ignore. Therefore, a compelling argument can be made for including neo-epitopes from these antigens as part of approaches to elicit tolerance. Obviously, carefully designed clinical trials will be necessary to validate approaches as both safe and effective and to identify epitopes that are suitable for eliciting tolerance either in subjects with T1D in general or in particular disease endotypes. Accompanying mechanistic studies should be employed to determine immune pathways that promote tolerance and to identify complementary approaches that can be combined to establish and reinforce tolerance.

## 5. Conclusions

T1D is an autoimmune disease in which the loss of tolerance to self-antigens including insulin, GAD65, IA-2, and ZnT8 leads to autoreactive T cell responses that promote the destruction of pancreatic *β* cells. Here we have outlined several new classes of non-genetically encoded epitopes and explained how these represent a significant challenge to central and peripheral tolerance mechanisms. Notably, these specificities (including hybrid insulin peptides, citrullinated and deamidated peptides, and unique splice variant junctions) are generated through processes that are intimately linked to the unique biology of *β* cells. The relative abundance of T cells that recognize these epitopes suggests that they make an important contribution to the progression of islet autoimmunity in T1D and provides motivation to include these specificities as possible targets for strategies to induce antigen specific tolerance.

## Figures and Tables

**Figure 1 biomedicines-09-00202-f001:**
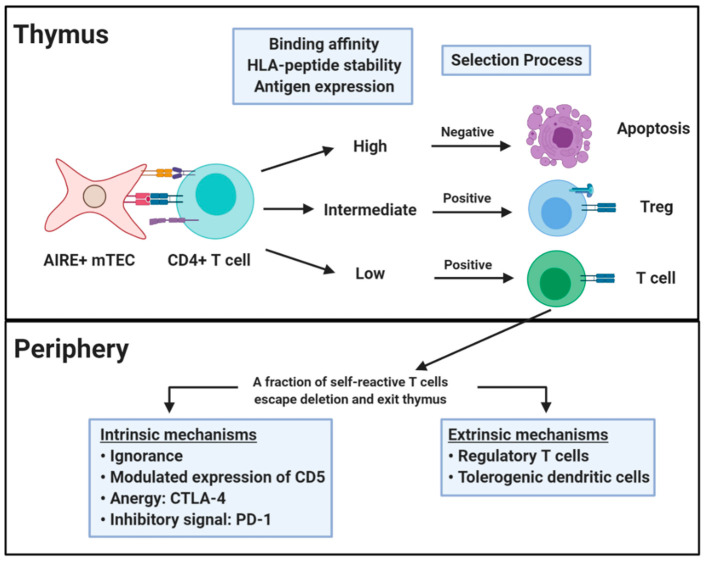
Central and peripheral tolerance. Central tolerance occurs in the thymus and relies on self-antigen expression on medullary thymic epithelial cells (mTEC) under regulation of the transcription factor AIRE. CD4^+^ T cells that recognize self-antigens in the context of HLA class II proteins undergo positive or negative selection. The selection process is based on the nature of the interaction between TCR and HLA-peptide complex. T cells that are positively selected go on to populate the periphery where they are controlled by peripheral tolerance mechanisms acting directly on the self-reactive T cells (intrinsic) or indirectly via additional cells (extrinsic). Figure created with BioRender.com.

**Figure 2 biomedicines-09-00202-f002:**
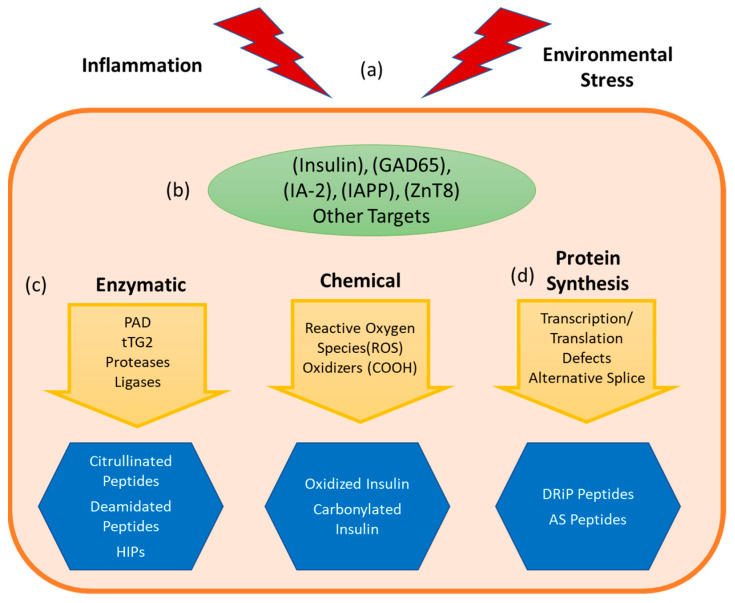
Formation of neo-antigens and epitopes. An undetermined stress event (**a**) combines with the beta cell’s unique metabolism and vulnerability to cellular and environmental stresses, leading to a cascade of deleterious events. Consequently, tissue specific proteins (**b**) are altered, leading to the formation of immunogenic neoepitopes (**c**) via enzymatic catalyzed pathways such as citrullination and deamidation, or biochemical reformation of reactive sidechains by processes such as oxidation. (**d**) Neoepitopes can also arise due to events during transcription and translation, producing alternatively spliced (AS) peptide junctions or defective ribosomal products (DRiPs). These neoepitopes can be presented by antigen presenting cells (APCs) resulting in autoreactive T cell responses.

**Table 1 biomedicines-09-00202-t001:** Tolerance induction strategies.

Technique	Administration Route ^3^	Effect	References
Direct peptide delivery	INH/NAS/ID	IL-10 secretion by ag-specific CD4^+^ T cells	[88,89]
p/MHC ^1^	IV	Induction of ag-specific Tr1-like ^4^ cells and regulatory network	[90]
p/MHC nanoparticle ^1^	INH/IT	Induction of tolerogenic APCs and reduced autoreactive T cell proliferation	[91,92]
DNA encoding self-peptide	IM	Reduced IFN-γ and altered co-stimulation.	[94]
Tolerogenic DCs ^2^	PO/ID	Improved regulatory/effector T cell ratio	[96,97]
Soluble antigen arrays	SC	Induction of IL-10 and exhausted-like T cells	[98]

^1^ p/MHC indicates HLA-peptide complexes ^2^ DCs indicates regulatory dendritic cells. ^3^ Administration routes are indicated using standard abbreviations. ^4^ Tr1 indicates CD4^+^ regulatory type 1-like T cells.

## Data Availability

Not applicable.

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
