# Peer review of "Non-Genetically Encoded Epitopes Are Relevant Targets in Autoimmune Diabetes"

_biomedicines, 2021, doi:10.3390/biomedicines9020202_

Round 1
Reviewer 1 Report
I have only few suggestions to make:
- The authors are suggested to include in the title of the review their focus on T1D (instead of autoimmunity in general).
- I would also suggest the subtitle “Oxidized and insulin” (line 183) be changed into “Oxidation and insulin”.
- The authors are recommended to recheck their manuscript for some grammatical and syntax errors throughout the text. A native English speaker could be helpful.
Author Response
Reviewer#1:
I have only few suggestions to make:
- The authors are suggested to include in the title of the review their focus on T1D (instead of autoimmunity in general).
<The title has been modified as suggested
- I would also suggest the subtitle “Oxidized and insulin” (line 183) be changed into “Oxidation and insulin”.
<The subtitle has been revised as recommended
- The authors are recommended to recheck their manuscript for some grammatical and syntax errors throughout the text. A native English speaker could be helpful.
<All four authors are native speakers. We have corrected the limited number of typos that were present in the article. Furthermore, we performed an extensive search for grammatical errors. As a result, we restructured a few sentences and added some commas.
Reviewer 2 Report
In their paper “Non-genetically Encoded Epitopes are Relevant Targets in Autoimmune Disease” Nguyen et al. describe the role of islet antigen and reactive T cells in beta cell destruction in T1D. The scientific findings are nicely reviewed, they are described clearly and concisely making the review easy to read and follow. I only have a few minor comments.
I would suggest changing the title to "....Targets in Autoimmune Diabetes" instead of "...Targets in Autoimmune Disease". I understand that most mechanisms described apply to other autoimmune diseases as well, but this review clearly focuses on T1D.
Please expand the section of T1D treatment options via induction of antigen specific tolerance.
Author Response
Reviewer#2:
In their paper “Non-genetically Encoded Epitopes are Relevant Targets in Autoimmune Disease” Nguyen et al. describe the role of islet antigen and reactive T cells in beta cell destruction in T1D. The scientific findings are nicely reviewed, they are described clearly and concisely making the review easy to read and follow. I only have a few minor comments.
I would suggest changing the title to "....Targets in Autoimmune Diabetes" instead of "...Targets in Autoimmune Disease". I understand that most mechanisms described apply to other autoimmune diseases as well, but this review clearly focuses on T1D.
<The title has been modified as suggested.
Please expand the section of T1D treatment options via induction of antigen specific tolerance.
<We have extensively expanded that the section on T1D treatment options as requested. For example, we now include one of the newest strategies (soluble antigen arrays). The paragraph now includes additional content throughout. In particular, we noticed and remedied the fact that we had not adequately discussed tolerogenic DCs.
Reviewer 3 Report
The authors present a nice summary on the role of non-genetically encoded epitopes in T1D. They analysed how tolerance is lost to insulin and other beta cell specific antigen and then described non-genetically encoded epitopes generated for example from post-translationally modified antigens and hybrid insulin peptides.
The authors conclude that CD4+ T cells that recognise non-genetically encoded epitopes are likely to make an important contribution to the progression of T1D and propose that these classes of epitopes should be considered as possible targets for strategies to induce antigen specific tolerance.
Overall, the narrative of the manuscript is good and the story is properly described. I think a few points should be addressed prior to publication.
-In the introduction the authors describe the relevance of auto-reactive T cells in T1D and the disease risk associated with susceptible HLA alleles. Subsequently they briefly describe the relevance of T cells response to non-genetically encoded neo-epitopes. Please expand the section relative to non-genetically encoded epitopes.
- Figure 2 and neo-epitope mechanisms implicated in T1D should be deeply described in section 3 (Non-genetically encoded neo-antigens and epitopes)
-lines 179-182: it would be better to cite few example and related specific research articles at this point.
- Multiple environmental factors are thought to play a role in the events leading to T1D. Are this factor also implicated in the generation of non-genetically encoded neo-antigens and epitopes? Please discusses it in the manuscript.
-The manuscript focuses on T1D. Please empathise it in the title.
Author Response
Reviewer#3:
The authors present a nice summary on the role of non-genetically encoded epitopes in T1D. They analysed how tolerance is lost to insulin and other beta cell specific antigen and then described non-genetically encoded epitopes generated for example from post-translationally modified antigens and hybrid insulin peptides.
The authors conclude that CD4+ T cells that recognise non-genetically encoded epitopes are likely to make an important contribution to the progression of T1D and propose that these classes of epitopes should be considered as possible targets for strategies to induce antigen specific tolerance.
Overall, the narrative of the manuscript is good and the story is properly described. I think a few points should be addressed prior to publication.
-In the introduction the authors describe the relevance of auto-reactive T cells in T1D and the disease risk associated with susceptible HLA alleles. Subsequently they briefly describe the relevance of T cells response to non-genetically encoded neo-epitopes. Please expand the section relative to non-genetically encoded epitopes.
<To preserve the intended organization of our article, we reserve some aspects of this topic for the sections and paragraphs that follow the introduction. However, to allay the concern that we did not adequately frame the topic in the introduction, we added the following sentences:
“As we delineate further in the sections that follow, the formation of non-genetically encoded antigens and epitopes represents a significant challenge to self-tolerance. Their formation represents a nexus between the natural secretory function of pancreatic β cells and inflammatory and environmental stresses. The result is the generation of self-proteins with non-templated sequence changes that alter their immunogenicity.”
- Figure 2 and neo-epitope mechanisms implicated in T1D should be deeply described in section 3 (Non-genetically encoded neo-antigens and epitopes)
<The text has been revised as requested to further incorporate Figure 2 into this section of the paper.
-lines 179-182: it would be better to cite few example and related specific research articles at this point.
<These lines look ahead to the ensuing paragraphs on insulin oxidation, hybrid insulin peptides, and insulin defective ribosomal products, all of which are well referenced. Out of deference to this concern we revised that sentence and added one additional sentence to the paragraph:
“However, decades of study make it clear that biological processes can alter the proinsulin transcript or modify proinsulin and insulin proteins at key amino acids, thereby generating non-genetically encoded insulin epitopes that pose a tolerogenic challenge (19). We will individually discuss the most important of these processes (insulin oxidation, formation of hybrid insulin peptides, and defective ribosomal products of proinsulin) in the subsections that follow.”
- Multiple environmental factors are thought to play a role in the events leading to T1D. Are this factor also implicated in the generation of non-genetically encoded neo-antigens and epitopes? Please discusses it in the manuscript.
<We had already mentioned the link between cellular stress and generation of non-genetically encoded neo-antigens and epitopes. We have revised the text as follows to also highlight the potential role of initiating environmental factors in that process:
“The environmental trigger that precipitates the loss of tolerance in T1D is unknown, but it is notable that processes shown to generate neo-sequences can be easily tied to suspected triggers of T1D. For example, enteroviral infection (e.g. with Coxsackie virus, which accelerates NOD diabetes) is a known trigger for inflammatory cytokines, such as interferon-γ (44). The anti-viral immune response is thought to synergize with pre-existing autoreactive T-cells to precipitate destruction of β cells (45). The secretion of pro-inflammatory cytokines by infiltrating immune cells (46, 47) sets off a cascade of events that includes HLA class I upregulation, expression of pro-apoptotic proteins, mitochondrial dysfunction and endoplasmic reticulum (ER) stress (48, 49). These events converge to elicit important changes to both the transcriptional profiles and the proteome of β cells (50-53). Beyond viral infection, there are other plausible triggers, including metabolic stress (54) and exposure to chemicals (55, 56). As will be discussed below, one common downstream component of these putative environmental triggers is increased cytosolic calcium, leading to increased activity of the peptidyl arginine deiminase (PAD) enzymes responsible for protein citrullination and tissue transglu-taminase (tTG) enzyme responsible for protein deamidation (40). Complementary mechanisms lead to the increased formation and presentation of splice variant pep-tides and DRiP derived peptides.”
-The manuscript focuses on T1D. Please empathise it in the title.
<The title has been modified as suggested.